# Learning about an exponential amount of conditional distributions

**Mohamed Ishmael Belghazi**[1,2]
ishmael.belghazi@gmail.com

**Maxime Oquab**[1]
qas@fb.com

**Yann Lecun**[1]
yann@fb.com

**David Lopez-Paz**[1]
dlp@fb.com

[1]Facebook AI Research, Paris, France
[2]Montréal Institute for Learning Algorithms, Montréal, Canada

## Abstract

We introduce the Neural Conditioner (NC), a self-supervised machine able to learn about all the conditional distributions of a random vector $X$. The NC is a function $NC(x \cdot a, a, r)$ that leverages adversarial training to match each conditional distribution $P(X_r | X_a = x_a)$. After training, the NC generalizes to sample conditional distributions never seen, including the joint distribution. The NC is also able to auto-encode examples, providing data representations useful for downstream classification tasks. In sum, the NC integrates different self-supervised tasks (each being the estimation of a conditional distribution) and levels of supervision (partially observed data) seamlessly into a single learning experience.

## 1 Introduction

Supervised learning estimates the conditional distribution of a target variable given values for a feature variable [63]. Supervised learning is the backbone to build state-of-the-art prediction models using large amounts of labeled data, with unprecedented success in domains spanning image classification, speech recognition, and language translation [35]. Unfortunately, collecting large amounts of labeled data is an expensive task painstakingly performed by humans (for instance, consider labeling the objects appearing in millions of images). If our ambition to transition from machine learning to artificial intelligence is to be met, we must build algorithms capable of learning effectively from inexpensive unlabeled data without human supervision (for instance, millions of unlabeled images). Furthermore, we are interested in the case where the available unlabeled data is partially observed. Thus, the goal of this paper is unsupervised learning, defined as understanding the underlying process generating some partially observed unlabeled data.

Currently, unsupervised learning strategies come in many flavors, including component analysis, clustering, energy modeling, and density estimation [23]. Each of these strategies targets the estimation of a particular statistic from high-dimensional data. For example, principal component analysis extracts a set of directions under which the data exhibits maximum variance [28]. However, powerful unsupervised learning should not commit to the estimation of a particular statistic from data, but extract general-purpose features useful for downstream tasks.

An emerging, more general strategy to unsupervised learning is the one of self-supervised learning [24, for instance]. The guiding principle behind self-supervised learning is to set up a supervised learning problem based on unlabeled data, such that solving that supervised learning problem leads to partial understanding about the data generating process [32]. More specifically, self-supervised learning algorithms transform the unlabeled data into one set of input features and one set of output

features. Then, a supervised learning model is trained to predict the output features from the input features. Finally, the trained model is later leveraged to solve subsequent learning tasks efficiently. As such, self-supervision turns unsupervised learning into the supervised learning problem of estimating the conditional expectation of the output features given the input features. A common example of a self-supervised problem is image in-painting. Here, the central patch of an image (output feature) is predicted from its surrounding pixel values (input feature), with the hope that learning to in-paint leads to the learning of non-trivial image features [50, 38]. Another example of a self-supervised learning problem extracts a pair of patches from one image as the input feature, and requests their relative position as the target output feature [10]. These examples hint one potential pitfall of "specialized" self-supervised learning algorithms: in order to learn a single conditional distribution from the many describing the data, it may be acceptable to throw away most of the information about the sought generative process, which in fact we would like to keep for subsequent learning tasks.

Thus, a general-purpose unsupervised learning machine should not commit to the estimation of a particular conditional distribution from data, but attempt to learn as much structure (i.e., interactions between variables) as possible. This is a daunting task, since joint distributions can be described in terms of an exponential amount of conditional distributions. Thus, learning the joint distribution, a problem usually associated to unsupervised learning, can be understood as analogous to an exponential amount of supervised learning problems. Our challenges do not end here. Being realistic, learning agents never observe the entire world. For instance, occlusions and camera movements hide portions of the world that we would otherwise observe. Therefore, we are interested in unsupervised learning algorithms able to learn about the structure of unlabeled data from partial observations.

In this paper, we address the task of unsupervised learning from partial data by introducing the Neural Conditioner (NC). In a nutshell, the NC is a function $\text{NC}(x \cdot a, a, r)$ that leverages adversarial training to match each conditional distribution $P(X_r | X_a = x_a)$. The set of available variables $a$, the set of requested variables $r$, and the set of available values $x \cdot a$ can be either determined by the pattern of missing values in data, or randomly by the self-supervised learning process. The set of available variables $a$ and the set of requested variables $r$ are not necessarily complementary, and index an exponential amount of conditional distributions (each associated to a single self-supervised learning problem). After trained, the NC generalizes to sample from conditional distributions never seen during training, including the joint distribution. Furthermore, trained NC's are also able to auto-encode examples, providing data representations useful for downstream classification tasks. Since the NC does not commit to a particular conditional distribution but attempts to learn a large amount of them, we argue that our model is a small step towards general-purpose unsupervised learning. Our contributions are as follows:

- We introduce the Neural Conditioner (NC) (Section 2), a method to perform unsupervised learning from partially observed data.

- We explain the multiple uses of NCs (Section 3), including the generation of conditional samples, unconditional samples, and feature extraction from partially observed data.

- We provide insights on how NCs work and should be regularized (Section 4).

- Throughout a variety of experiments on synthetic and image data, we show the efficacy of NCs in generation and prediction tasks (Sections 5 and 7).

## 2   The Neural Conditioner (NC)

Consider the dataset $(x_1, \ldots, x_n)$, where each $x_i \in \mathbb{R}^d$ is an identically and independently distributed (iid) example drawn from some joint probability distribution $P(X)$. Without any further information, we could consider $O(3^d)$ different prediction problems about the random vector $X$, where each prediction problem partitions the coordinates $x_i$ into features, targets, or unobserved variables. We may index this exponential amount of supervised learning problems using binary vectors of *available features* $a \in \{0, 1\}^d$ and *requested features* $r \in \{0, 1\}^d$. In statistical terms, a pair of available and requested vectors $(r, a)$ instantiates the supervised learning problem of estimating the conditional distribution $P(X_r | X_a = x_a)$, where $x_r = (x_i : r_i = 1)$, and $x_a = (x_i : a_i = 1)$.

By making use of the notations above, we can design a single supervised learning problem to estimate all the conditional distributions contained in the random vector $X$. Since learning algorithms are often designed to deal with inputs and outputs with a fixed number of dimensions, we will consider

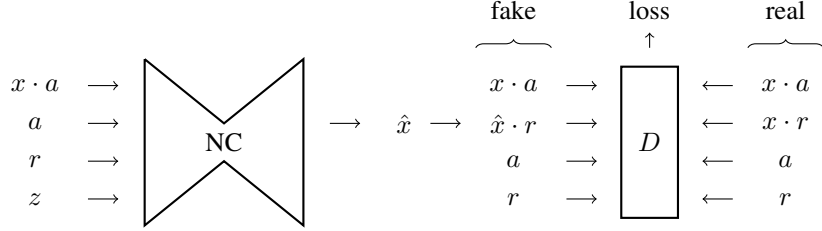

Figure 1: The proposed NC, where data $x \sim P(X)$, available/requested masks $a, r \sim P(a, r)$, and noise $z \sim \mathcal{N}(0, I)$.

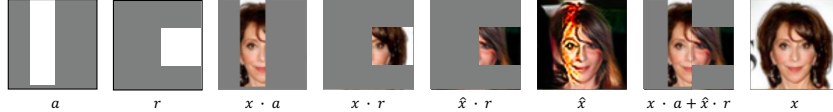

Figure 2: Example of masks and masked images. NC learns to predict $x \cdot r$ from $x \cdot a$.

the augmented supervised learning problem of mapping the feature vector $(x \cdot a, a, r)$ into the target vector $x \cdot r$, where the operation "$\cdot$" denotes entry-wise multiplication. In short, our goal is to learn a Neural Conditioner (NC) producing samples:

$$\hat{x} \sim \text{NC}(x \cdot a, a, r) : \hat{x}_r \sim P(X_r | X_a = x_a) \ \forall \ (x, a, r).$$

The previous equation manifests the ambition of NC to model the entire conditional distribution $P(X_r | X_a = x_a)$ when given a triplet $(x, a, r)$. Therefore, given the dataset $(x_1, \ldots, x_n)$, learning a NC translates into minimizing the distance between the estimated conditional distributions $\text{NC}(x \cdot a, a, r)$ and the true conditional distributions $P(X_r | X_a = x_a)$, based on their samples. In particular, we will follow recent advances in implicit generative modeling, and implement NC training using tools from generative adversarial networks [18]. Other alternatives to train NCs would include maximum mean discrepancy metrics [21], energy distances [61], or variational inference [31]. If the practitioner is only interested in recovering a particular statistic from the exponentially many conditional distributions (e.g. the conditional means), training a NC with a scoring rule $D$ for such statistic (e.g. the mean squared error loss) would suffice.

Training a NC is an iterative process involving six steps, illustrated in Figures 1 and 2:

1. A data sample $x$ is drawn from $P(X)$.

2. Available and requested masks $(r, a)$ are drawn from some data-defined or user-defined distribution $P(R, A)$. These masks are not necessarily complementary, enabling the existence of unobserved (neither requested or observed) variables. If a coordinate equals to one in both $r$ and $a$, we zero it at the requested mask.

3. A noise vector $z$ is sampled from an external source of noise with distribution $P(Z)$.

4. A sample is generated as $\hat{x} = \text{NC}(x \cdot a, a, r, z)$.

5. A discriminator $D$ provides the final scalar objective function by distinguishing between data samples (scored as $D(x \cdot r, x \cdot a, a, r)$) and generated samples (scored as $D(\hat{x} \cdot r, x \cdot a, a, r)$).

6. The NC parameters are updated to minimize the objective function, while the parameters of the discriminator are updated to maximize it, following adversarial training [18].

Mathematically, our general objective function is:

$$\min_{\text{NC}} \max_{D} \ \mathop{\mathbb{E}}_{x, a, r} \ \log D(x \cdot r, x \cdot a, a, r) + \mathop{\mathbb{E}}_{x, a, r, z} \ \log(1 - D(\text{NC}(x \cdot a, a, r, z) \cdot r, x \cdot a, a, r)). \quad (1)$$

## 3 Using NCs

Once trained, one NC serves many purposes. The most direct use is perhaps the *multimodal prediction of any subset of variables given any subset of variables*. More specifically, a NC is able to leverage

any partially observed vector $x_a$ to predict about any partially requested vector $x_r$. Importantly, the combination of test values, available, and requested masks $(x, a, r)$ could be novel and never seen during training. Since NCs leverage an external source of noise $z$ to make their predictions, NCs provide a conditional distribution for each triplet $(x, a, r)$.

Two special cases of masks deserve special attention. First, properly regularized NCs are able to *compress and reconstruct samples* when provided with the full requested mask $r = 1$ and the full available mask $a = 1$. This turns NCs into autoencoders able to *extract feature representations of data*, as well as allowing *latent interpolations between pairs of examples*. Second, when provided with the full requested mask $r = 1$ and the empty available mask $a = 0$, NCs are able to *generate full samples from the data joint distribution* $P(X)$, even in the case when the training never provided the NC with this mask combination, as our experiments verify.

NCs are able to seamlessly *deal with missing features and/or labels during both training and testing time*. Such "missingness" of features and labels can be real (as given by incomplete or unlabeled examples) or simulated by designing an appropriate distribution of masks $P(A, R)$. This blurs the lines that often separate unsupervised, semi-supervised, and supervised learning, integrating all types of data and supervision into a new learning paradigm.

Finally, a trained NC can be used to understand relations between variables, for instance by using a complete test vector $x$ and querying different available and requested masks. The strongest relations between variables can also be analyzed in terms of gradients with respect to $(a, r)$.

## 4 Understanding NCs

To better understand how NCs work, this section describes i) how NCs look like in the Gaussian case, ii) what the optimal discriminator minimizes, iii) the relationship between NC training and the usual reconstruction error minimized by auto-encoders, and iv) some regularization techniques.

### 4.1 The Gaussian case

Let us consider the case where the data joint distribution is a Gaussian $P(X) = \mathcal{N}(\mu, \Sigma)$. Then, the closed-form expression of the conditional distribution implied by any triplet $(x, a, r)$ is $P(X_r | X_a = x_a) = \mathcal{N}(\mu_{r|a}, \Sigma_{r|a})$, where $\mu_{r|a} = \mu_r + \Sigma_{ra} \Sigma_{aa}^{-1}(x_a - \mu_a)$, and $\Sigma_{r|a} = \Sigma_{rr} - \Sigma_{ra} \Sigma_{aa}^{-1} \Sigma_{ar}$.

The previous expressions highlight an interesting fact: even in the case of Gaussian distributions, computing the conditional moments implied by $(x, a, r)$ is a non-linear operation. When fixing $(a, r) = (a_0, r_0)$, learning the conditional distribution implied by triplets $(x, a_0, r_0)$ can be understood as linear heteroencoding [54].

The motivation behind self-supervised learning is that learning about a conditional distribution is an effective way to learn about the joint distribution. In part, this is because learning conditional distributions allows to deploy the powerful machinery of supervised learning. To formalize this, we consider the amount of information contained in a probability distribution in terms of its differential entropy. Then, we show that learning conditional distributions is easier than learning joint distributions, where "difficult" is measured in terms of how much information is to be learned. This argument can be made by considering the chain rule of the differential entropy [9]: $h(X) = \sum_{i=1}^{d} h(X_i | X_1, \ldots, X_{i-1})$, where, in the case of partitioning $X = (X_a, X_r)$, we have: $h(X) = h(X_r | X_a) + h(X_a)$. The previous shows that $h(X_r | X_a) \leq h(X_r)$, where equality is achieved if and only if $X_a$ and $X_r$ are independent. This reveals a "blessing of structure" of sorts: to reduce the difficulty of learning about a joint distribution, we should construct self-supervised learning problems associated to conditional distributions between highly coupled blocks of input and output features. Indeed, if all of our variables are independent, self-supervised learning is hopeless. For the case of a $d$-dimensional Gaussian with covariance matrix $\Sigma$, the differential entropy can be stated in terms of the covariance function: $h(\Sigma) = \frac{d}{2}(1 + \log(2\pi)) + \frac{1}{2}\log(|\Sigma|)$. which allows to choose good self-supervised learning problems based on the log-determinant of empirical covariances.

A successful evolution from single self-supervised learning problems to NCs rests on the existence of relationships between different conditional distributions. More formally, the success of NCs relies on assuming a smooth landscape of conditionals. If smoothness across conditional distributions is satisfied, learning about some conditional distribution should inform us about other, perhaps never

seen, conditionals. This is akin to supervised learning algorithms relying on smoothness properties of the function to be learned. For NCs we do not consider the smoothness of a single function, but the smoothness of the "conditioning operator" $C_x(a, r) = \text{NC}(x \cdot a, a, r)$. The smoothness of this conditioning operator is related to the smoothness of the covariance operator studied in kernel embeddings of distributions [45].

## 4.2 Training objective, NC's point of view

This section considers the following question: what is the objective function minimized by NC? In particular we are interested in the intriguing fact of how NCs is able to complete and reconstruct samples, when the discriminator is never presented with pairs of real and generated requested variables. First, consider the "augmented" data and model $\theta$ joint distributions

$$P(X_a, X_r, A, R) = q(X_r|X_a, A, R)p(X_a, A, R),$$
$$P_\theta(X_a, X_r, A, R) = q_\theta(X_r|X_a, A, R)p(X_a, A, R).$$

Next, consider the negative log-likelihood $L(x_a, a, r) = -\mathbb{E}_q \log q_\theta$ and its expectation $L = -\mathbb{E}_P \log q_\theta$. Then,

$$L(X_a, A, R) = -\mathbb{E}_q \log \left( q_\theta \cdot \frac{q}{q} \right) = -\mathbb{E}_q \left\{ \log \frac{q_\theta}{q} + \log q \right\} = \int -q \log q - \int q \log \frac{q_\theta}{q}.$$

Integrating wrt $p(X_a, A, R)$, we see that NCs minimize:

$$L = D_{KL}(P \parallel P_\theta) + H(X_R|X_A) = D_{KL}(P \parallel P_\theta) - I(X_A, X_R) + H(X_R)$$
$$= D_{KL}(P \parallel P_\theta) - I(X_A, X_R) + H(X_A).$$

Where $H$ stands for (conditional) entropy and $I$ for mutual information. Following [18], assuming an optimal discriminator and a NC globally minimizing (1) we have that $P = P_\theta$, $D_{KL}(P \parallel P_\theta) = 0$, and thus $L = H(X_R|X_A)$.

We summarize the previous results as follows. If a NC is able to match the distributions $(P, P_\theta)$, there will be a residual reconstruction error of $H(X_R \mid X_A)$. Thus, if $X_A$ and $X_R$ are independent, such residual reconstruction error reduces to $H(X_R)$. This can happen if $A = 0$, or if $X_A$ holds no information about $X_R$. Moreover the reconstruction error is a decreasing function of the amount of information that $X_A$ holds about $X_R$.

## 4.3 Regularization

We found, during our experiments, gradient based regularization on the discriminator to be crucial. Following [53] we augment the discriminator's loss with the expected gradient with respect to the inputs for both the positive and negative examples; Less succintly, we add $\frac{1}{2}(\mathbb{E}[\|\nabla D(X_A, X_R, A, R)\|^2] + \mathbb{E}[\|\nabla D(X_A, \hat{X}_R, A, R)\|^2])$ to the discriminator's loss.

For NC to generalize to unobserved conditional distributions and prevent memorizing the observed ones, we have found that regularization of the latent space to be essential. In information theoretic terms, we would like to control the mutual information between $X_A$ and $Z := enc(X_A, \epsilon)$. One could use a variational approximation of the conditional entropy [1] or an adversarial approach [3]. The former requires an encoder with tractable conditional density (e.g. Gaussian), the latter, while allowing general encoders, introduces an additional training loop in the algorithm. We opt for another approach by controlling the encoder's Lipschitz constant using one-sided spectral normalization [42].

## 5 Experiments on Gaussian data

We train a single NC to model all the conditionals of a three-dimensional Gaussian distribution. Given that in this example we know that the data generating process is fully determined by the first two moments, we train two versions of NCs: one that uses moment-matching, and one that uses our full adversarial training pipeline. Both strategies train NC given minibatches of triplets $(x, a, r)$ observed from the same Gaussian distribution. This allows us to better understand the impact of adversarial training when dealing with NCs. For these experiments, both the discriminator and the NC have 2 hidden layers of 64 units each, and ReLU non-linearities. We regularize the latent space

Figure 3: Illustration of the NC on a three-dimensional Gaussian dataset. We show a) one-dimensional conditional estimation, b) two-dimensional conditional estimation, and c,d) the representation of the conditional distributions in the hidden space.

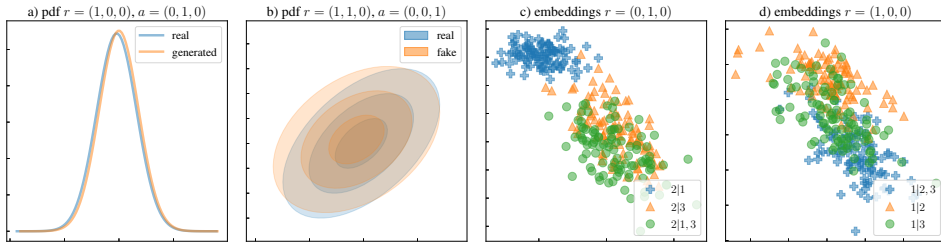

Table 1: Average error norms $\|\theta_{r|a} - \hat{\theta}_{r|a}\|$ in the task of estimating the conditional moments $\theta_{r|a} = (\mu_{r|a}, \Sigma_{r|a})$ of Gaussian data. We show results for Moment-Matching (MM) and the full Adversarial Training (AT). VAEAC only supports complementary masks (some results are NA).

| $a$ | $r$ | NC (MM) | NC (AT) | VAEAC |
|---|---|---|---|---|
| (1, 0, 0) | (0, 0, 1) | $.09 \pm .06$ | $.10 \pm .05$ | NA |
| | (0, 1, 0) | $.10 \pm .04$ | $.07 \pm .03$ | NA |
| | (0, 1, 1) | $.67 \pm .05$ | $.13 \pm .04$ | $.68 \pm .03$ |
| (0, 1, 0) | (0, 0, 1) | $.16 \pm .03$ | $.08 \pm .05$ | NA |
| | (1, 0, 0) | $.20 \pm .05$ | $.05 \pm .03$ | NA |
| | (1, 0, 1) | $.28 \pm .07$ | $.14 \pm .06$ | $.73 \pm .03$ |
| (0, 0, 1) | (0, 1, 0) | $.13 \pm .06$ | $.11 \pm .07$ | NA |
| | (1, 0, 0) | $.08 \pm .05$ | $.09 \pm .05$ | NA |
| | (1, 1, 0) | $.29 \pm .03$ | $.17 \pm .03$ | $.71 \pm .06$ |
| (1, 0, 1) | (0, 1, 0) | $.22 \pm .07$ | $.11 \pm .07$ | $.50 \pm .04$ |
| (1, 1, 0) | (0, 0, 1) | $.15 \pm .08$ | $.08 \pm .05$ | $.43 \pm .03$ |
| (0, 1, 1) | (1, 0, 0) | $.27 \pm .09$ | $.15 \pm .07$ | $.35 \pm .05$ |

| | | NC conditioning | |
|---|---|---|---|
| | | $\varnothing$ | $(a, r)$ |
| discriminator | $\varnothing$ | 0.12 | 0.17 |
| conditioning | $(a, r)$ | 0.15 | 0.07 |

of the NC using one-sided spectral normalization [43] We train the networks for $10,000$ updates, with a batch-size of $512$, and the Adam optimizer with a learning rate of $10^{-4}$, $\beta_1 = 0.5$, and $\beta_2 = 0.999$. The training set contains $10^4$ fixed samples sampled from a Gaussian with mean $(2, 4, 6)$ and covariance $((1, 0.5, 0.25), (0.5, 1, 0), (0.25, 0, 1))$.

Figure 3 illustrates the capabilities of NC to perform one-dimensional and two-dimensional conditional distribution estimation. We also show the embeddings of the conditional distributions as given by the bottleneck of NC. These show a higher dependence for variables that are more tightly coupled. Table 1 shows the error on the conditional parameter estimation for the NC (both using moment matching and adversarial training) as well as the VAEAC [26], a VAE-based analog to the NC. Finally, Table 1 (right) shows the importance of conditioning both the discriminator and NC on both available and requested masks.

## 6    Missing data Imputation

In order to quantitatively evaluate NC's ability to construct representations of the joint distributions from data with missing observations, we consider data imputation tasks on three UCI datasets [37]. We compare to GAIN [66] and use the same empirical setup for the sake of consistency. Note that while [66] augments the adversarial loss with an euclidean reconstruction error NC does not. Table 2 shows the normalized root mean squared error of the imputed missing data on the test set.

## 7    Experiments on image data

We train NCs on SVHN and CelebA. We use rectangular $a, r$ masks spanning between 10% and 50% of the images. We evaluate our setup in several ways. First qualitatively: generating full samples (using the never seen mask configuration $a = 0, r = 1$, Fig 4) and reconstructing samples (Figures 5 for denoising and 6 for inpainting). These experiments share the goal of showing that our model is

Table 2: RMSE of missing data Imputations on the test set. Experiments were repeated five times.

| Algorithm | Spam | Letter | Credit |
|---|---|---|---|
| MICE[55] | $.0699 \pm .0010$ | $.1537 \pm .0006$ | $.2585 \pm .0011$ |
| MissForest[60] | $.0553 \pm .0013$ | $.1605 \pm .0004$ | $.1976 \pm .0015$ |
| Matrix[40] | $.0542 \pm .0006$ | $.1442 \pm .0006$ | $.2602 \pm .0073$ |
| Auto-encoder[17] | $.0670 \pm .0030$ | $.1351 \pm .0009$ | $.2388 \pm .0005$ |
| EM[14] | $.0712 \pm .0012$ | $.1563 \pm .0012$ | $.2604 \pm .0015$ |
| GAIN w/o $l_2$[66] | $.0672 \pm .0036$ | $.1586 \pm .0024$ | $.2533 \pm .048$ |
| GAIN[66] | $.0513 \pm .0016$ | $.1198 \pm .005$ | $.1858 \pm .0010$ |
| VAEAC[26] | $.0552 \pm .0020$ | $.1115 \pm .0010$ | $.1523 \pm .0020$ |
| NC | $\mathbf{.0486 \pm .0010}$ | $\mathbf{.0851 \pm .0020}$ | $\mathbf{.1276 \pm .0020}$ |

able to generalize to conditional distributions not observed during training. Second, we evaluate our models quantitatively: that is, their ability to provide useful features for downstream classification tasks (see Table 3). Our results show that NC-based figures systematically outperform state-of-art hand-crafted features, while being competitive with deep unsupervised features.

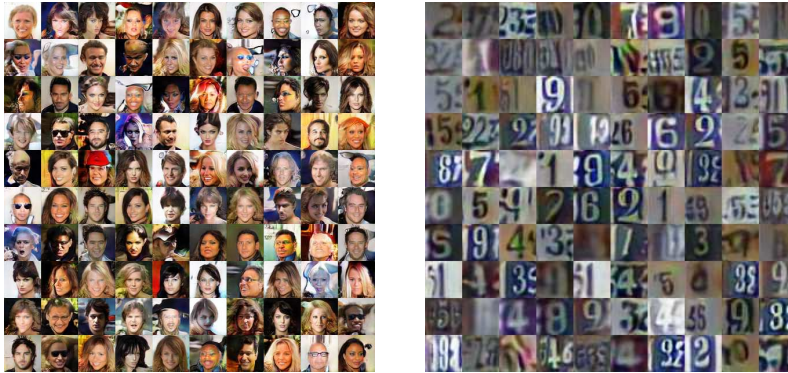

Figure 4: SVHN and CelebA samples. The model never observed a complete sample in training.

Figures 5 and 6 show samples and in-paintings using masks configurations unobserved during training to illustrate that our model is able to generalize to conditional distributions and construct representation of the data solely through partial observation. Figure 4 shows samples from the joint distribution ($a = 0, r = 1$), even though these masks were never observed during training.

### 7.0.1 Feature extraction

**SVHN**   As a feature extraction procedure, we retrieve the latent code created by the PAE while feeding an image in *compress and reconstruct* mode ($a = r = 1$). Then, we use a linear SVM to assess the quality of the extracted encoding, and show in Table 3 that our approach is competitive with deep unsupervised feature extractors.

**CelebA**   The multimodality presented by the CelebA attributes provides an ideal test mode to quantify our model ability to construct a global understanding out of local and partial observations. Following [6, 39], we train 40 linear SVMs on learned representations extracted from the encoder using full available and requested masks ($a = r = 1$) on the CelebA validation set. We measure the performance on the test set. As in [6, 25, 29], we report the *balanced accuracy* in order to evaluate the attribute prediction performance. Please note that our model was trained trained on entirely unsupervised data and masking configurations unobserved during training. Attribute labels were only used to train the linear SVM classifiers.

Table 3: Test errors on SVHN (left), and test accuracies on CelebA (right).

| Model | Test error |
|---|---|
| KNN | 77.93 |
| TSVM | 66.55 |
| VAE (M1 + M2) [30] | 36.02 |
| DCGAN + L2-SVM [51] | 22.18 |
| ALI + L2-SVM [13] | $19.14 \pm 0.50$ |
| VAEAC [26] | $57.89 \pm 1.0$ |
| NC (L2-SVM) (ours) | $\mathbf{17.12 \pm 0.59}$ |

| Model | Mean | Stdv |
|---|---|---|
| Triplet-kNN [57] | 71.55 | 12.61 |
| PANDA [67] | 76.95 | 13.33 |
| Anet [39] | 79.56 | 12.17 |
| LMLE-kNN [25] | **83.83** | **12.33** |
| VAE [31] | 73.30 | 9.65 |
| ALI [13] | 73.88 | 10.16 |
| HALI [4] | **83.75** | **8.96** |
| VAEAC [27] | 66.06 | 6.98 |
| NC (Ours) | **82.21** | **7.63** |

# 8   Related work

Self-supervised learning is an emerging technique for unsupervised learning. Perhaps the earliest example of self-supervised learning is auto-encoding [2, 24], which in the language of NCs amounts to full available and requested masks. Auto-encoders evolved into more sophisticated variants such as denoising auto-encoders [64], a family of models including NC. Recent trends in generative adversarial networks [18] are yet another example of self-supervised training. The connection between auto-encoders and generative adversarial training was first instantiated by [34]. Auto-regressive models [5] such as the masked autoencoder [15], neural autoregressive distribution estimators [33, 62], and Pixel RNNs [47] are other examples of casting unsupervised learning using a simple self-supervision strategy: order the variables, and then predict each of them using the previous.

Moving further, the task of unsupervised learning with partially observed data was also considered by others, often in terms of estimating transition operators [20, 7, 58]. Generative adversarial imputation nets [66] considered the case of learning missing feature predictions using adversarial training. In a different thread of research, the literature in kernel mean embeddings [59, 36, 45] is an early consideration of the problem of learning distributions. Concerning applications, self-supervised learning was pioneered by word embeddings [41]. In the image domain, self-supervised setups include image in-painting [50], colorization [68], clustering [8], de-rotation [16], and patch reordering [10, 46]. In the video domain, common self-supervised strategies include enforcing similar feature representations for nearby frames [44, 19, 65], or predicting ambient sound statistics from video frames [48]. These applications yield representations useful for downstream tasks, including classification [8], multi-task learning [11], and RL [49]. Finally, the most similar piece of literature to our research is the concurrent work on VAE with Arbitrary Conditioning, or VAEAC [26]. The VAEAC is proposed as a fast alternative to the also related universal marginalizer [12]. Similarly to our setup, the VAEAC augments a VAE with a mask of requested variables; the complimentary set of variables is provided as the available information for prediction. Our work extends VAEAC by employing adversarial training to obtain better sample quality and features for downstream tasks. To sustain these claims, a comparison between NC and VAEAC was performed in Section 7. As commonly assumed in VAE-like architectures, the conditional encoding and decoding distributions are assumed Gaussian, which may not be a good fit for complex multimodal data such as natural images. The VAEAC work was mainly applied to the problem of feature imputation. Here we hope to provide a more holistic perspective on the uses of NCs, including feature extraction and semi-supervised learning.

# 9   Conclusion

We presented the Neural Conditioner (NC), an adversarially-learned neural network able to learn about the exponentially many conditional distributions describing some partially observed unlabeled data. Once trained, one NC serves many purposes: sampling from (unseen) conditional distributions to perform multimodal prediction, sampling from the (unseen) joint distribution, and auto-encode (partially observed) data to extract data representations useful for (semi-supervised) downstream tasks. Neural Conditioner blurs the lines that often separate unsupervised, semi-supervised, and supervised learning, integrating all types of data and supervision into a holistic learning paradigm.

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
