[Supplementary Material]

# Appendices

## A   Performance degradation on train/test masks mismatch

We analyze the performance of NC as a function of the mismatch between train and test masks.
To this end, we report the RMSE between $X_R$ and $\hat{X}_R$ on the UCI/Letter dataset.
We consider four different scenarios: (i) masks observed during training applied on train data, (ii) masks observed during training applied on test data, (iii) masks unobserved during training applied on train data, and (iv) masks unobserved during training applied on test data.
Table A that there is minimal performance degradation when using unobserved masks and/or examples at test time showing the model's ability to generalize accross conditional distributions and examples.

Table 4: RMSE on UCI/Letter using train/test data/masks.

| Masks/Data | Train | Test |
|---|---|---|
| Train | $.0891 \pm .001$ | $.0896 \pm .001$ |
| Test | $.0897 \pm .001$ | $.0901 \pm .001$ |

# B  Experimental details

**UCI datasets**  We use a two hidden layer MLP with ReLU activations and 1024 hidden units for both the projector and the discriminator. The projector's final layer is a sigmoid. Both networks are initialized following [56].

Available and requested masks are encoded as Radamecher random variables. Furthermore, we normalize the masks energy before feeding them to either the projector or the discriminator.

We train our model for 20000 iterations using Adam [31] with $\lambda = 1e - 4$, $\beta_1 = 0$ and $\beta_2 0.999$ and a batch size of 128.

**SVHN**  The discriminator closely follows the architecture presented in [22]. The generator is U-net[52] architecture where both the encoder and decoder consist of residual blocks. The details are provided in the tables below.

| Discriminator | | | | | |
|---|---|---|---|---|---|
| | Kernel size | Stride | Padding | BN | Output dim |
| Residual Block | 3 | 2 | 1 | $\times$ | $128 \times 16 \times 16$ |
| Residual Block | 3 | 2 | 1 | $\times$ | $128 \times 8 \times 8$ |
| Residual Block | 3 | 1 | 1 | $\times$ | $128 \times 8 \times 8$ |
| Residual Block | 3 | 1 | 1 | $\times$ | $128 \times 8 \times 8$ |
| Global mean pooling | $-$ | $-$ | $-$ | $\times$ | $128 \times 1 \times 1$ |
| Conv | 1 | 1 | 0 | $\times$ | $1 \times 1 \times 1$ |

Table 5: Discriminator's architecture (SVHN)

| Encoder (Projector) | | | | | |
|---|---|---|---|---|---|
| | Kernel size | Stride | Padding | BN | Output dim |
| Residual Block | 3 | 2 | 1 | $\times$ | $128 \times 16 \times 16$ |
| Residual Block | 3 | 2 | 1 | $\checkmark$ | $128 \times 8 \times 8$ |
| Residual Block | 3 | 2 | 1 | $\checkmark$ | $128 \times 4 \times 4$ |
| Conv | 4 | 1 | 0 | $\times$ | $128 \times 1 \times 1$ |

Table 6: Encoder architecture (SVHN)

| Decoder (Projector) | | | | | | |
|---|---|---|---|---|---|---|
| | Kernel size | Stride | Padding | BN | Upsampling | Output dim |
| Conv | 1 | 1 | 0 | $\times$ | $\times$ | $128 \times 4 \times 4$ |
| Residual Block | 3 | 1 | 1 | $\checkmark$ | Nearest | $128 \times 8 \times 8$ |
| Residual Block | 3 | 1 | 1 | $\checkmark$ | Nearest | $128 \times 16 \times 16$ |
| Residual Block | 3 | 1 | 1 | $\checkmark$ | Nearest | $128 \times 32 \times 32$ |
| Conv | 3 | 1 | 1 | $\times$ | $\times$ | $3 \times 32 \times 32$ |

We require available and requested masks to be non-overlapping and randomly cover 10% to 50% of the total image area. We encode the masks as Radamecher random variables and normalize their energy.

All the networks are initialized following [56]. We train our model for 100000 iterations using Adam [31] with $\lambda = 2e - 4$, $\beta_1 = 0$ and $\beta_2 0.9$ and a batch size of 128.

**CelebA**  The discriminator closely follows the architecture presented in [22]. The generator is U-net[52] architecture where both the encoder and decoder consist of residual blocks. The details are provided in the table below.

| Discriminator | | | | | |
|---|---|---|---|---|---|
| | Kernel size | Stride | Padding | BN | Output dim |
| Residual Block | 3 | 2 | 1 | $\times$ | $64 \times 32 \times 32$ |
| Residual Block | 3 | 2 | 1 | $\times$ | $128 \times 16 \times 16$ |
| Residual Block | 3 | 2 | 1 | $\times$ | $256 \times 8 \times 8$ |
| Residual Block | 3 | 1 | 1 | $\times$ | $512 \times 4 \times 4$ |
| Global mean pooling | $-$ | $-$ | $-$ | $\times$ | $512 \times 1 \times 1$ |
| Conv | 1 | 1 | 0 | $\times$ | $1 \times 1 \times 1$ |

Table 7: Discriminator's architecture (CelebA)

| Encoder (Projector) | | | | | |
|---|---|---|---|---|---|
| | Kernel size | Stride | Padding | BN | Output dim |
| Residual Block | 3 | 2 | 1 | $\times$ | $64 \times 32 \times 32$ |
| Residual Block | 3 | 2 | 1 | $\checkmark$ | $128 \times 16 \times 16$ |
| Residual Block | 3 | 2 | 1 | $\checkmark$ | $256 \times 8 \times 8$ |
| Residual Block | 3 | 2 | 1 | $\checkmark$ | $512 \times 4 \times 4$ |
| Conv | 4 | 1 | 0 | $\times$ | $128 \times 1 \times 1$ |

Table 8: Encoder architecture (CelebA)

| Decoder (Projector) | | | | | | |
|---|---|---|---|---|---|---|
| | Kernel size | Stride | Padding | BN | Upsampling | Output dim |
| Conv | 1 | 1 | 0 | $\times$ | $\times$ | $512 \times 4 \times 4$ |
| Residual Block | 3 | 1 | 1 | $\checkmark$ | Nearest | $256 \times 8 \times 8$ |
| Residual Block | 3 | 1 | 1 | $\checkmark$ | Nearest | $128 \times 16 \times 16$ |
| Residual Block | 3 | 1 | 1 | $\checkmark$ | Nearest | $64 \times 32 \times 32$ |
| Residual Block | 3 | 1 | 1 | $\checkmark$ | Nearest | $64 \times 64 \times 64$ |
| Conv | 3 | 1 | 1 | $\times$ | $\times$ | $3 \times 64 \times 64$ |

We require available and requested masks to be non-overlapping and randomly cover 10% to 50% of the total image area. We encode the masks as Radamecher random variables and normalize their energy.

All the networks are initialized following [56]. We train our model for 100000 iterations using Adam [31] with $\lambda = 2e - 4$, $\beta_1 = 0$ and $\beta_2 0.9$ and a batch size of 128.

# C  Qualitative results

Figure 5: Denoising SVHN images corrupted with 50% missing pixels using a model *trained on square masks*.

Figure 6: In-painting SVHN images using masks of size and shapes *not seen during training*.

Figure 7: Predicting partially-observed CelebA images. From left to right: $x \cdot a$, $x \cdot r$, $\hat{x} \cdot r$, $\hat{x}$, $(x \cdot a + \hat{x} \cdot r)$, $x$. Saturation patterns happen only for pixels where $a = 1$.