[Reviews · NeurIPS 2019]

Reviewer 1



# Response to rebuttal After reading the author response, as well as the other reviews, I think that the authors have addressed most of the concerns I had raised in my review, in spite of the limited time and space available, in what I consider to be an excellent rebuttal. With those concerns now out of the way, I have decided to raise my score to 8 (clear accept). # Summary This paper proposes a novel deep generative model that aims to learn not only the joint distribution of the data but rather all $3^{d} - 1$ conditional distributions that could be derived from the joint, where $d$ is the number of individual features to be modelled. To this end, the authors extend generative adversarial networks (GANs) by: 1) Defining the generator to be a function $NC(x \cdot a, a, r, z)$ of i) prior noise $z$; ii) binary masks $a$ and $r$ indicating the available (observed) and requested (latents to be inferred) features; and iii) observed features themselves $\left\{x_{i} \mid a_{i} = 1 \right\}$, encoded as a fixed-dimensional object via masking, i.e., $x \cdot a$. In other words, the generator can be considered to be a conditional GAN (cGAN) with the auxiliary information being the observed features and the available and requested masks. 2) Defining the discriminator to be a function $D(x \cdot r, x \cdot a, a, r)$ not only of the requested features $\left\{x_{i} \mid r_{i} = 1 \right\}$, again encoded via masking as $x \cdot r$, but also of all the auxiliary information $x \cdot a$, $a$ and $r$. The setup for the discriminator can thus also be considered to mimic cGANs. 3) Generalizing the training procedure by randomly generating available/requested masks for each minibatch from some distribution defined a priori. The performance of the proposed approach is evaluated by means of i) toy experiments on a synthetic Gaussian-distributed dataset with $d=3$ features; ii) experiments assessing the ability of the model to impute missing features on three UCI datasets; and iii) experiments showing that the model can generate and reconstruct samples, as well as learning unsupervised representations useful for downstream classification tasks, using SVHN and CelebA as benchmark datasets. # High-Level Assessment The core idea behind the manuscript, that of developing a model able to simultaneously capture all possible conditional distributions of the data, has already been recently addressed by the Variational Autoencoder with Arbitrary Conditioning (VAEAC) [21]. However, this manuscript makes original contributions to the problem statement in at least three different dimensions: i) An approach complementary to [21] is proposed, relying on adversarial training instead of variational autoencoding. ii) The proposed method models all $3^{d} - 1$ conditional distributions while the approach in [21] requires the available and requested features to be complementary and thus models “only” $2^{d}$ conditional distributions. iii) Empirically, the authors report that the proposed approach outperforms [21] in its ability to model toy data (Table 1) and to learn useful representations for downstream classification tasks in an unsupervised manner (Table 3, CelebA). However, I believe the current version of the manuscript has some shortcomings involving (i) lack of clarity / rigor in the theoretical analysis in Sections 4.2 and 4.3; (ii) insufficient experimental results, specially regarding the comparison with the VAEAC and the characterization of certain limitations of the proposed approach; and (iii) insufficient clarity regarding low-level aspects of the experimental setup, which I believe would make it difficult to reproduce the paper based on the text alone. While I believe these issues should be resolved prior to publication, I consider the approach presented in the manuscript to be technically sound and novel, the empirical results to be promising and the problem statement itself to be of great importance to the field. Therefore, I would be glad to increase my score if the aforementioned issues are addressed or clarified in case some of them were misguided. # Major Points 1. In Section 4.2, a theoretical analysis of the discriminator objective is shown to justify the specific form chosen for the discriminator function and generally to provide insight about the way the method operates. However, while the objective shown in Equation (1) resembles the original GAN objective introduced in [14], it appears that the authors “switched” to an alternative criterion for the discriminator in Section 4.2, corresponding to a lower bound on the KL divergence derived from the Donsker-Varadhan representation [3]. However, the manuscript does not discuss or justify this apparent inconsistency. 2. Likewise, in Section 4.3 a theoretical analysis complementary to Section 4.2 is presented, studying the training objective from the point of view of the generator. However, I do not immediately understand why what I believe to be the core argument of this section holds true: “Next, consider the negative log-likelihood $L(x_{a}, a, r) = - \mathbb{E}_{q} \log q_{\theta}$ and its expectation $L = - \mathbb{E}_{P} \log q_{\theta}$. Recall that the latter expectation is the objective function minimized by generators in the usual non-saturating GAN objective [14], such as it happens in NC.” In particular, given that the non-saturating GAN objective is usually given by $-\mathbb{E}_{x \sim P_{g}} \log D(x)$, I do not not see why that would equal $L = - \mathbb{E}_{P} \log q_{\theta}$, since the proposed expectation is taken with respect to the data distribution $P$ instead of the generative distribution $P_{\theta}$ and, based on the analysis in the preceding section, the optimal discriminator $D^{*}$ would not equal $q_{\theta}$. 3. The fact that, unlike the VAEAC, the proposed approach also aims to model conditional distributions for which some features have been marginalized makes it important to consider the issue of consistency. In particular, there appears to be nothing in the model to enforce that, for example, if $\left\{i \mid r^{\prime}_{i} = 1 \right\} \subset \left\{i \mid r_{i} = 1 \right\}$, the implicit conditional distribution $P(X_{r^{\prime}} \mid X_{a} = x_{a})$ captured by the model would agree with the alternative implicit conditional distribution $\widetilde{P}(X_{r^{\prime}} \mid X_{a} = x_{a})$ obtained by marginalizing the features $\left\{i \mid r_{i} = 1, r^{\prime}_{i} = 0 \right\}$ in $P(X_{r} \mid X_{a} = x_{a})$. While I believe solving this limitation could certainly be left for future work, it would be desirable to at least study the extent to which the learnt conditional distributions are mutually consistent empirically. 4. Despite being clearly the most relevant baseline, the comparison with VAEAC in the experiments is not exhaustive. In particular, it is not included in the imputation experiments described in Section 6, the qualitative sample quality experiments in Section 7 nor the unsupervised representation learning experiments for the SVHN dataset in Section 7.0.1. 5. Key low-level details are missing from the paper, making it difficult to reproduce the results from the manuscript alone. Perhaps most crucially, the authors do not discuss the specific implementation of the generator $NC(x \cdot a, a, r, z)$ and discriminator $D(x \cdot r, x \cdot a, a, r)$ functions and how the different inputs are combined in practice. Other aspects of the experimental setup could also be clearer, for example, in Section 5 it is not described which distribution is used to generate masks during training nor how exactly the conditional embeddings shown in Figure 3 are computed, and descriptions of the model architectures are largely missing in Section 7. # Minor Points 6. The results shown in Tables 1 and 3 (SVHN) would benefit from reporting error bars. 7. While the experiments in Section 7 suggest that the model is indeed able to generalize to unseen available/requested mask configurations, this could have been studied in a more systematic way, for example, by investigating how different performance metrics degrade as a function of the mismatch between the train and test mask configurations. It could also have been interesting to measure the extent to which the performance of the proposed approach would improve in Table 3 if it had also been trained using the mask configurations for the autoencoding setting. # Typos Line 120: full requested mask $r = 0$ -> full requested mask $r = 1$. Lines 190 - 191: $\int q \log q$ is missing a minus sign. Line 232: NC ability -> NC’s ability Line 232: representation -> representations Line 233: observation -> observations Line 233: consider a data imputation tasks -> consider data imputation tasks Line 233: space missing before citation [31] Line 234: space missing before citation [53] Line 235: reference to Table 2 seems to be broken Line 254: extractor -> extractors Line 257: representation is duplicated Line 258: avaialable -> available

Reviewer 2



This work proposes a self-supervised learning framework called the Neural Conditioner (NC) which attempts to model all possible conditional distributions P(X_r|X_a) where X_r and X_a denote partial observations (i.e., of a subset of variables) of a random vector X. The NC is learned using adversarial training where a generator estimates the subset of variables to be recovered X_r given the available subset of variables X_a, and a discriminator tries to discriminate between the estimates and true observations. A trained NC is shown to be capable of sampling from conditional distributions that were unseen at training time, including the joint distribution, and also learns features that are useful for downstream tasks. While the idea of modeling the conditional distribution X_r|X_a has been explored in prior work (Eg: VAEAC), this work differs by (i) using adversarial training, which enables generating more realistic output and (ii) showing that the model is useful for many different tasks such as estimating conditionals (including unseen conditional distributions) ad producing features useful for downstream tasks. The paper is well written in general. Ideas are well described, and modeling choices look reasonable. I appreciate the authors providing a lot of intuition about the different components of the approach, performing experiments on a toy setup where things are interpretable, and showing comprehensive results on different tasks as well comparing against a strong baseline from prior work (VAEAC).

Reviewer 3



The proposed method is somewhat interesting for data imputation. It seems to be a good choice if one wants to imputation missing data at random locations. This is also empirically validated in the experiments. In addition to data imputation, the authors proposed several other motivations for their proposed method. However, I have several significant concerns The claims in the introduction are questionable. For example, “unsupervised learning in general is an exponential amount of supervised learning problems” seem overly audacious, and needs justification. There are many similar unconvincing statements that should either be justified or downplayed. There is certainly much more to be desired from unsupervised learning, such as learning meaningful features that benefit downstream tasks, or data visualization. It is not clear how the proposed approach benefit these common accepted use cases of unsupervised learning. I am not convinced by the arguments of the benefit of learning the conditional distribution. For example, why should the entropy have anything to do with our ability to model a distribution? (line 154) A standard Gaussian distribution can have higher (Shannon) entropy than the complex distribution over natural images, but it is strange to claim that the latter is easier to learn/model. I'm not sure I get the significance of the theoretical analysis (section 4.2-4.3). For example, for section 4.2. Does this provide any additional insight compared with standard minimax distribution analysis of GANs? The notation is also sloppy (notation such as C_b is not even defined). The same holds true for section 4.3. It is not clear what all the probabilities are, are they conditional probabilities? Joint probabilities? The notation leaves a lot of room for guessing. In addition, since the objective (eq 1) is stanfard GAN and not f-GAN, it is not clear where the KL comes from. Therefore, I am not confident that these analysis are correct (but maybe it is because I didn't parse the notation correctly). Experiments: The experiments are okay, the authors show that the proposed approach is able to impute missing data. The difference from GAIN is minor, but the proposed method performs slightly better. In fact, the contribution of a superior imputation method is the major argument I am judging this paper on (because of my previously mentioned concerns on the other claimed contributions), but it is only briefly explored with simple experiments showing small net improvement. The experiments on semi-supervised learning is unsurprisingly un-par with other generic deep generative model based methods. Writing: The non-technical part of the paper is reasonably well written and clear. However, the theory sections are harder to understand because of ambiguous/unclear notation. Small points: Typo: line 80 a \in {0, 1} -> a \in {0, 1}^n -------------- After rebuttal The authors provide additional experiments on imputation. Combining with the original experiments, I am fairly convinced that the proposed approach is good for data imputation. I'm still not convinced that "learning good features" is a significant benefit of the proposed method. More experimental validation (downstream task performance and visualization) would be necessary to prove good feature learning performance. For example, in Table 1 of the rebuttal, the performance seem far from state-of-the-art methods. Nonetheless I am happy to improve my score by 1 because of the convincing imputation results.

[Author Response · NeurIPS 2019]

**Motivation of the method**   Following Reviewer #3, we clarify the motivation behind NC. We emphasize our goal of learning features from unsupervised data, useful for downstream tasks. This is the main point validated throughout our experiments. We show that our paradigm of learning many conditional distributions of the data allows the extraction of these unsupervised features from incomplete data, as well as arbitrary data imputations (inpaintings).

**General changes to the manuscript**   Following Reviewer's #1 suggestion, we included in the Appendix our experimental protocols, architectures, and optimization parameter grids for all methods. We also added error bars to Tables 1 and 3 from the manuscript. Following Reviewer's #3 suggestion, we tone down our claims about unsupervised learning.

**Regarding Sections 4.2 and 4.3**   We apologize for the confusion brought by using the Donsker-Varadhan lower bound as an objective for the discriminator. As pointed out by Reviewer #1, we acknowledge that the statement from lines 188-190 is misleading. Our intent was to reason about optimal discriminators. Given an optimal discriminator, the optimal NC minimizes the Jensen-Shannon divergence between $p_{\theta^*}(x_r \mid x_a, a, r)$ and $p(x_r \mid x_a, a, r)$, where $p_\theta$ and $p$ represent the model and data distributions, respectively. Consequently, at optimality we have that $D_{KL}(p||p_{\theta^*}) = 0$, and thus the negative log-likelihood is equal to $H(X_R|X_A)$. Then, the more information $X_A$ holds about $X_R$, the lower the negative log-likelihood. Following Reviewer's #1 and #3 remarks, we replace the Donsker-Varadhan lower bound by one in terms of the Jensen-Shannon divergence, merging Sections 4.2 and 4.3, removing the misleading statement from lines 188-190, making clear that our reasoning follows for optimal discriminators and NC's, and making the variables on which the different distributions depend explicit. We thank the reviewers for their careful reading.

**Regarding conditional distributions consistency**   We thank Reviewer #1 for bringing this subtle point to our attention. We now provide a proof about the consistency of conditional and marginalized densities in our Appendix. The proof sketch goes as follows: if we assume that the data has support on a compact set $\Omega$, and that the NC is trained to optimality, then, writing $\lambda$ for the Lebesgue measure on $\Omega$, we can show that $D_{JS}(p_\theta(x_S \mid x_A)||\int p_\theta(x_S, x_{R-S} \mid x_A)\lambda(dx_{R-S}))$ is small by leveraging the triangular inequality of the distance on probability measure on $\Omega$ defined by the square root of the Jensen-Shannon divergence. We then use Jensen's inequality with uniform weights $\frac{1}{\lambda(supp(X_{R-S}))}$ to bound the distance between the model and data marginalized distributions by the integral of the distance between the model and data conditional distributions. We leave the theoretical analysis for the case for non-compact supports to future work. Moreover, we illustrate this consistency between conditional and marginalized densities empirically, here in the Figure on the right.

**Better empirical comparison against VAEAC**   As suggested, we improve our empirical comparison against VAEAC and update the manuscript with the results shown here in Table 1 (Left, semi-supervised learning results on SVHN), and Table 1 (Middle, missing data imputation on three UCI datasets).

**Improved performances on the missing data imputation task**   As requested by Reviewer #3, Table 1 (Middle) shows substantially improved missing data imputations results for our model. This results were obtained after fixing a bug in our code.

**Performance degradation depending on train/test masks mismatch**   We follow Reviewer's #1 suggestion to analyze the performance of NC as a function of the mismatch between train and test masks. To this end, we report the RMSE between $X_R$ and $\hat{X}_R$ on the UCI/Letter dataset. We consider four different scenarios: (i) masks observed during training applied on train data, (ii) masks observed during training applied on test data, (iii) masks unobserved during training applied on train data, and (iv) masks unobserved during training applied on test data. Results in Table 1 (Right).

| Algorithm | Test error (%) | Algorithm | Spam | Letter | Credit | Masks/Data | Train | Test |
|---|---|---|---|---|---|---|---|---|
| VAEAC | $57.89 \pm 1.01$ | GAIN | $.0513 \pm .002$ | $.1198 \pm .005$ | $.1858 \pm .001$ | Train | $.0891 \pm .001$ | $.0896 \pm .001$ |
| NC | $\mathbf{17.2 \pm 0.59}$ | VAEAC | $.0552 \pm .002$ | $.1115 \pm .001$ | $.1523 \pm .002$ | Test | $.0897 \pm .001$ | $.0901 \pm .001$ |
| | | NC | $\mathbf{.0486 \pm .001}$ | $\mathbf{.0851 \pm .002}$ | $\mathbf{.1276 \pm .002}$ | | | |

Table 1: **(Left)** Semi-supervised learning on SVHN using 1000 labels. **(Middle)** RMSE for missing data imputation on UCI datasets. **(Right)** RMSE on UCI/Letter using train/test data/masks.

[Meta-Review · NeurIPS 2019]

This paper proposes a novel neural mechanism to model many conditionals. The reviewers agreed it was novel and showed value.